# Transient and Recurrent Vision Loss in a High-Altitude Porter from Pakistan on a Polish Winter Karakoram Expedition

**DOI:** 10.3390/ijerph182212204

**Published:** 2021-11-20

**Authors:** Robert K. Szymczak, Magdalena Sawicka

**Affiliations:** 1Department of Emergency Medicine, Faculty of Health Sciences, Medical University of Gdansk, Mariana Smoluchowskiego 17, 80-214 Gdansk, Poland; robert.szymczak@gumed.edu.pl; 2Department of Neurology, Faculty of Medicine, Medical University of Gdansk, Mariana Smoluchowskiego 17, 80-214 Gdansk, Poland

**Keywords:** altitude, hypoxia, visual aura, vision loss, migraine without pain, altitude induced migraine

## Abstract

Visual sensations appear in most migraine auras, but binocular blindness is uncommon. We described a case of multiple transient losses of vision in a man on a winter expedition to K2. His symptoms were later diagnosed as recurrent visual auras without pain. Sojourns at altitude can induce migraine attack; therefore, susceptible individuals should avoid factors that might provoke migraines at high altitude, such as improper acclimatization, dehydration and an inadequate sleep regime.

## 1. Introduction

Migraine is one of the most common diseases that might be preceded by an aura, i.e., a focal neurological impairment occurring before headache. Visual sensations appear in more than 90% of migraine auras [1], but binocular blindness is uncommon. Sometimes, migraine aura occurs without being followed by a headache, but that is a rare clinical phenomenon [2]. Many factors trigger migraine. Hypoxia is also among them and might induce migraine attacks [3]. 

We described a case involving multiple occurrences of transient vision loss in a man on a winter expedition to K2 whose reported symptoms were later diagnosed as recurrent visual auras without pain.

This case is unique. Binocular blindness among migraineurs is a rare occurrence, and this uncommon form of migraine aura was repeated seven times, every time without a subsequent headache, which is also infrequent. The history happened while climbing the second highest mountain in the world. Very few detailed case reports of migraine aura without headache have been presented in adults. 

## 2. Materials and Methods

### Case Description 

A 45 year old high altitude porter from Pakistan on a winter Polish expedition to K2 (Figure 1) suffered transient vision loss seven times at 5100 m above sea level (Figure 2). He was an experienced porter accustomed to high altitudes and had never previously experienced these visual disturbances. This man did not drink alcohol and had never smoked. He had no chronic diseases and did not take medications for any condition. His mother suffered from recurrent severe headaches, which were undiagnosed because of the low level of medical care in the area they inhabit. She reported experiencing a throbbing pain on one side of her head accompanied by nausea, vomiting, and sensitivity to light and sound without focal neurological impairments before the headache. This pain was so severe that it interfered with her daily activities. No other family member was ill. 

Each instance of the porter’s blindness lasted about 20 min, began gradually and started at rest once a day for seven consecutive days. Each incident displayed identical symptoms. The accompanying sign was fatigue, but no headache. These binocular disturbances resolved spontaneously and completely, and were unaccompanied by pain in the eyeballs. He did not suffer nausea, vomiting, limb weakness or speech disorder during the attacks. He also did not report photophobia after blindness onset or previous to vision loss. The porter used eye protection against ultraviolet radiation and was well acclimatized to the 5100 m altitude of the base camp. This man reported no earlier symptoms that might have suggested altitude illness. During the described events and when the porter was asymptomatic, his heart rate and blood pressure were within normal limits and his level of oxygen saturation at the base camp was 85–88%.

## 3. Results

### Diagnostic Assessment

The porter consulted a doctor from Poland via satellite links who recommended that he leave the expedition and accompany a passing trekking group downhill before going to a hospital in Islamabad. There, fundoscopy with ophthalmological consultation, electrocardiography, echocardiography, routine electroencephalography and blood tests (a full blood count, erythrocyte sedimentation rate, platelets, creatinine, blood urea nitrogen, glucose, electrolytes, transaminases, lipids and coagulation profile) were conducted. None of these diagnosed any significant abnormalities. The computed tomography of his head (Figure 3) with angiography revealed no infarction, intracranial hemorrhage or mass lesion; the arterial and venous vasculature appeared normal, with no stenosis. No pathologies were found in the carotid and vertebral Doppler ultrasonography. The neurological examination was unremarkable. The patient consulted several experts in altitude medicine and neurology from the Department of Adult Neurology of the Medical University of Gdańsk (Poland), the Institute for Altitude Medicine in Ridgway (United States) and the Centre for Mountain and Neurology Medicine in Aosta Valley (Italy). 

Finally, after diagnostics, it was stated that the patient’s episodes of visual impairment corresponded with a recurrent migraine aura and were the equivalent of altitude-induced migraine. The porter did not continue his participation in this expedition.

## 4. Discussion

Though headache is the essence of migraine, migraine aura sometimes occurs without subsequent pain [2], but isolated aura, as a migraine equivalent, is a rare clinical phenomenon. The incidence of a typical aura without headache in migraine patients is about 3% in women and about 1% in men [4]. It can strike at any age and without a previous migraine history. Normally, the symptoms of aura are fully reversible and do not leave permanent neurological deficits [5], and diagnostic tests reveal nothing abnormal.

The visual aura might also manifest as a temporary loss of vision [6], though binocular blindness among migraineurs is a rare occurrence. Temporary binocular blindness usually presents in people with migraine as a single, isolated, totally reversible episode and is not related to other symptoms of aura [7]. 

The diagnosis of migraine (aura) should be considered in any patient at altitude who presents with focal neurological impairment, with or without headache. Whereas the possibility of a transient ischemic attack (TIA) or stroke should be considered as a priority, the diagnosis of a migraine aura should not be neglected, even when someone does not have a pain phase. 

Due to the fact that the presented man was in the mountains, it should be taken into consideration that hypoxia, dehydration, hemoconcentration, low temperature and immobilization from forced inactivity after exposure to a high altitude might have led to a hypercoagulable state and have predisposed a person to thromboembolic events, including ischemic stroke/TIA [8,9]. Focal visual symptoms of aura often mimic stroke, and in clinical practice migraine aura could be difficult to distinguish from transient ischemic attacks and stroke [10]. The quality of visual aura symptoms is an important feature that might help to distinguish migraine aura from life-threatening conditions [10]. 

Brain tumors might also become suddenly symptomatic upon exposure to a high altitude [11,12]. Moreover, paroxysmal visual disturbances could be symptoms of epilepsy and represent epileptic seizures arising from the occipital lobe, though nonconvulsive epilepsy with visual symptoms is an uncommon clinical phenomenon.

The fact that the problem was binocular, transient and recurrent ruled out most eye disorders.

Though the etiology of migraine is not fully understood, environmental factors, including exposure to high altitudes, have been linked to migraine development. Insufficient oxygen supply (hypoxia) leads to many changes in the brain and neurological disabilities, including migraine attacks. For people prone to migraines, decreasing oxygen concentrations in the air can trigger migraine [13], and this porter’s case confirms that hypoxia might be involved in migraine pathophysiology, but the exact mechanism remains unknown. Possible pathophysiological mechanisms include hypoxia-induced cortical spreading depression or the release of the nitric oxide neuromodulator, which plays a role in cerebral blood flow regulation and in nociception, interacting with the calcitonin gene-related peptide (vasodilating neuropeptide) [14]. An association with mitochondrial energy metabolism (metabolic failure) is also considered [13]. The mechanisms behind the migraine-inducing effect of hypoxia need further investigation.

## 5. Conclusions

Sojourns at altitude might induce a migraine attack [15] (with or without an aura, though a migraine with aura is more common at higher altitudes than at sea level) or aggravate previous migraine headaches—their frequency, duration and the severity of symptoms. Migraine (aura) episodes can also start for the first time at higher altitudes, as in the case we described. Susceptible individuals should therefore avoid other factors that might provoke migraines in the mountains, such as improper acclimatization, dehydration, an inadequate sleep regime, some food and drinks, or stress [16]. To summarize the above, reducing other triggers at the time of ascent might be helpful in the prevention of migraine attacks.

## Figures and Tables

**Figure 1 ijerph-18-12204-f001:**
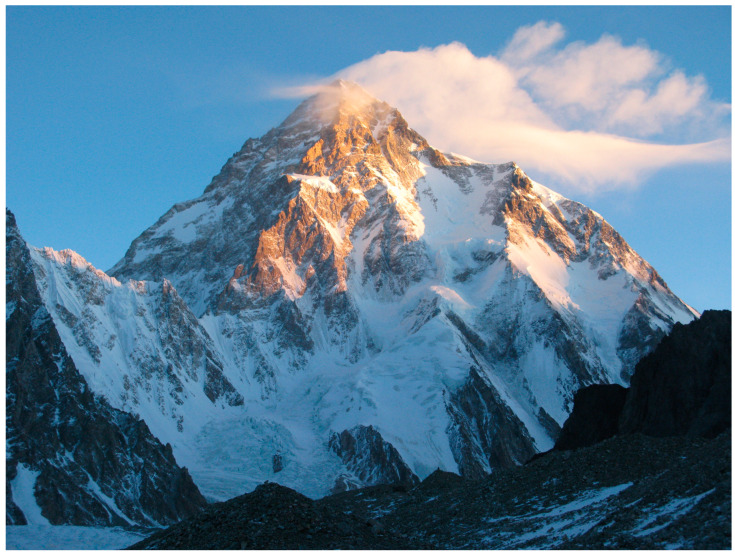
K2 in Karakoram, photo from a Polish winter expedition.

**Figure 2 ijerph-18-12204-f002:**
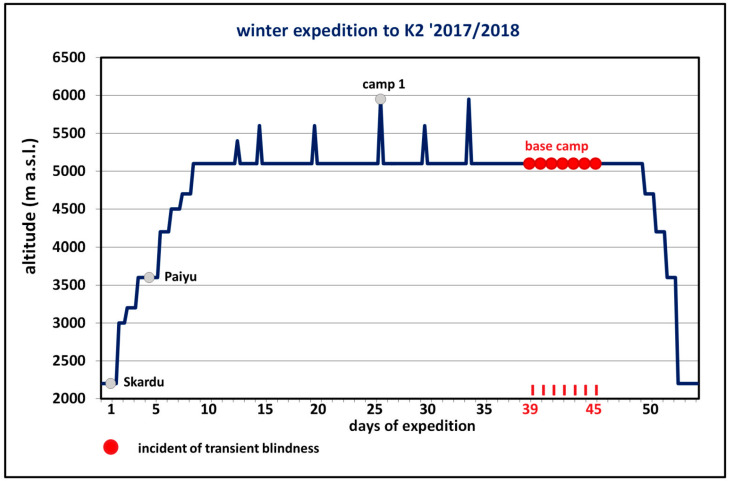
Acclimatization plan when conquering K2 until the occurrence of transient binocular blindness at the 5100 m altitude of the base camp during the Polish national winter expedition in 2017/2018.

**Figure 3 ijerph-18-12204-f003:**
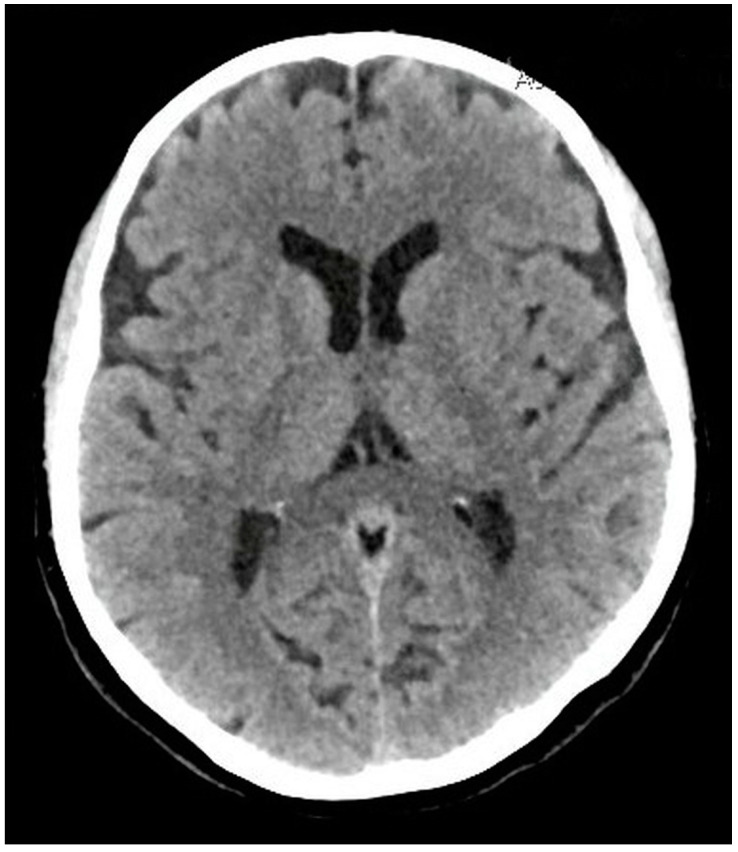
The computed tomography of the patient’s head did not reveal any abnormalities responsible for his symptoms.

## Data Availability

Data sharing not applicable.

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
