# Peer review of "Transient and Recurrent Vision Loss in a High-Altitude Porter from Pakistan on a Polish Winter Karakoram Expedition"

_ijerph, 2021, doi:10.3390/ijerph182212204_

Round 1
Reviewer 1 Report
Summary:
The paper entitled “Transient and recurrent vision loss in a high-altitude porter 2 from Pakistan on a Polish winter Karakoram expedition” by Szymczak and Sawicka discusses a case report of a porter on K2 expedition who has experienced periods of transient blindness seven times. Exploring the background of the porter there seems to be no apparent reason including his training and family history. His diagnostic assessment also suggests no abnormalities in fundoscopy, electrocardiography, echocardiography among other tests. This paper reports a new case of transient blindness, a form of migraine possibly induced by higher altitude.
Comments and Suggestions for Authors
- There are no citations in the introduction, would be useful to include some references.
- The case description needs to be slightly reframed where it is the better first to report the details of the porter, his background, and prior training to go on K2 expedition and then talk about the case with his symptoms in the order of their occurrence.
- For citation [6] may be another recent citation could be found.
Author Response
We have included citations in the Introduction as suggested. We have also reframed the case description by first describing the porter’s details, including his medical history, climbing background and family history, and then presenting the main case with symptoms. Family history of migraine is the most potent and consistent risk factor for migraine.
We have added a citation from 2021 on the manifestation of intracranial lesions at high altitude (Speidel, V.; Purrucker, J.C.; Klobučníková, K.; Manifestation of Intracranial Lesions at High Altitude: Case Report and Review of the Literature. High Alt Med Biol. 2021, 22, 87-89. doi: 10.1089/ham.2020.0223).
Reviewer 2 Report
The Authors present a case report on the occurrence of multiple visual loss disturbances diagnosed as aura without migraine in high-altitude porters during a winter expedition in Karakoram. The case report is well described, I would suggest to implement the discussion on the differential diagnosis of this symptomatology to be considered in that context as red flag. In addition, the discussion on the relationship between hypoxia and migraine (the aura should be considered even if without pain phase as a migraine) should be better structured.
Please find the following papers when you revise your manuscript.
PMID: 31146673
PMID: 26917583
Author Response
Reviewer: I would suggest to implement the discussion on the differential diagnosis of this symptomatology to be considered in that context as red flag.
We have presented the differential diagnosis of the presented symptomatology in Lines 111-123. Based on the suggested literature, we have added information about the importance of the quality of visual aura symptoms, which might help to distinguish migraine aura from life- threatening conditions.
Reviewer: In addition, the discussion on the relationship between hypoxia and migraine (the aura should be considered even if without pain phase as a migraine) should be better structured.
Basing on the suggested literature, we restructured our discussion on the relationship between hypoxia and migraine.
Please find the following papers when you revise your manuscript.
PMID: 31146673
PMID: 26917583
We have included these publications in our Discussion.
Reviewer 3 Report
This case report describes a case of a male patient who presented with multiple cases of binocular visual disturbances during a winter expedition at very high altitudes. This case reports an apparently rare phenomenon whereby migraine aura occurred without pain.
My main question is why this is particularly interesting to report? This does not come across clearly in the report. Is the focus on the binocular temporary blindness as a sympton?
Issue 1: As above: The introduction is brief, and while it mentions the novelty of the case but more explanation is required about why this is an interesting case to report.
Issue 2 : Citations are needed for all the statements in the introduction L20-25 citations
The case is comprehensively described and included a background of events during the seven incidents, health and family background assessments.
Issue 3 : It is unclear if this was the porter’s first trip at high altitudes, or if they had previously been at high altitude but not experienced the visual disturbances.
Issue 5 : In connection to his mother who suffered from recurrent severe headaches the authors state that “However, her headaches looked like a migraine”. Please rephrase based on the symptoms she experienced.
Issue 6: As with the introduction, there are additional citations needed within the discussion i.e Lines 84-88
Issue 7: The statement of informed consent appears to be incomplete
Author Response
Issue 1: As above: The introduction is brief, and while it mentions the novelty of the case but more explanation is required about why this is an interesting case to report.
We have improved the Introduction by including a paragraph explaining the novelty of the presented case and presenting reasons why it is interesting to report.
Issue 2 : Citations are needed for all the statements in the introduction L20-25 citations
We have added appropriate references.
The case is comprehensively described and included a background of events during the seven incidents, health and family background assessments.
Issue 3 : It is unclear if this was the porter’s first trip at high altitudes, or if they had previously been at high altitude but not experienced the visual disturbances.
We have added information about the porter’s previous climbing and medical history.
Issue 5 : In connection to his mother who suffered from recurrent severe headaches the authors state that “However, her headaches looked like a migraine”. Please rephrase based on the symptoms she experienced.
We have rephrased the section about the symptoms experienced by the patient’s mother.
Issue 6: As with the introduction, there are additional citations needed within the discussion i.e Lines 84-88.
We have added appropriate references.
Issue 7: The statement of informed consent appears to be incomplete
We have corrected the typing error in the Informed Consent Statement.